# Immunogenicity Following Administration of BNT162b2 and Ad26.COV2.S COVID-19 Vaccines in the Pregnant Population during the Third Trimester

**DOI:** 10.3390/v14020307

**Published:** 2022-02-02

**Authors:** Ioana Mihaela Citu, Cosmin Citu, Florin Gorun, Ioan Sas, Larisa Tomescu, Radu Neamtu, Andrei Motoc, Oana Maria Gorun, Bogdan Burlea, Felix Bratosin, Daniel Malita

**Affiliations:** 1Department of Internal Medicine I, “Victor Babes” University of Medicine and Pharmacy Timisoara, 300041 Timisoara, Romania; citu.ioana@umft.ro; 2Department of Obstetrics and Gynecology, “Victor Babes” University of Medicine and Pharmacy Timisoara, 300041 Timisoara, Romania; gorun.florin@umft.ro (F.G.); sasioan56@yahoo.com (I.S.); tomescu.larisa@umft.ro (L.T.); radu.neamtu@umft.ro (R.N.); 3Department of Anatomy and Embryology, “Victor Babes” University of Medicine and Pharmacy Timisoara, 300041 Timisoara, Romania; amotoc@umft.ro; 4Department of Obstetrics and Gynecology, Municipal Emergency Clinical Hospital Timisoara, 300202 Timisoara, Romania; oanabalan@hotmail.com (O.M.G.); bogdanburlea@yahoo.com (B.B.); 5Methodological and Infectious Diseases Research Center, Department of Infectious Diseases, “Victor Babes” University of Medicine and Pharmacy, 300041 Timisoara, Romania; felix.bratosin7@gmail.com; 6Department of Radiology, “Victor Babes” University of Medicine and Pharmacy Timisoara, 300041 Timisoara, Romania; malita.daniel@umft.ro

**Keywords:** SARS-CoV-2, COVID-19, pregnancy vaccination, BNT162b2, Ad26.COV2.S

## Abstract

Globally, COVID-19 vaccines are currently being used to prevent transmission and to reduce morbidity and death associated with SARS-CoV-2 infection. Current research reveals that vaccines such as BNT162b2 and Ad26.COV2.S are highly immunogenic and have high short-term effectiveness for most of the known viral variants. Clinical trials showed satisfying results in the general population, but the reluctance in testing and vaccinating pregnant women left this category with little evidence regarding the safety, efficacy, and immunogenicity following COVID-19 vaccination. With the worldwide incidence of COVID-19 remaining high and the possibility of new transmissible SARS-CoV-2 mutations, data on vaccination effectiveness and antibody dynamics in pregnant patients are critical for determining the need for special care or further booster doses. An observational study was developed to evaluate pregnant women receiving the complete COVID-19 vaccination scheme using the BNT162b2 and Ad26.COV2.S, and determine pregnancy-related outcomes in the mothers and their newborns, as well as determining adverse events after vaccination and immunogenicity of vaccines during four months. There were no abnormal findings in pregnancy and newborn characteristics comparing vaccinated versus unvaccinated pregnant women. COVID-19 seropositive pregnant women had significantly higher spike antibody titers than seronegative patients with similar characteristics, although they were more likely to develop fever and lymphadenopathy following vaccination. The same group of pregnant women showed no statistically significant differences in antibody titers during a 4-month period when compared with case-matched non-pregnant women. The BNT162b2 and Ad26.COV2.S vaccines are safe to administer during the third trimester of pregnancy, while their safety, efficacy, and immunogenicity remain similar to those of the general population.

## 1. Introduction

SARS-CoV-2 is a new coronavirus that has been discovered as the cause of coronavirus disease 2019 (COVID-19), causing significant morbidity and more than 5 million deaths around the world as a result of the illness (at the time of writing) [1]. According to accumulating data, pregnant women are more likely than non-pregnant women to develop COVID-19-related problems, which may include the requirement for invasive ventilation, admission to an intensive care unit, and death [2,3].

The COVID-19 vaccines, especially mRNA vaccines against SARS-CoV-2 such as BNT162b2 Pfizer and mRNA-1273 Moderna, were approved by the Food and Drug Administration (FDA) in late 2020 [4,5], initiating a global vaccination campaign where the vaccines were set to be administered in two doses. Later, a non-inferiority single-dose Ad26.COV2.S Johnson&Johnson vaccine, using a different technology, was approved by FDA. A significant number of people were evaluated in phase 3 randomized controlled trials, and both vaccinations were shown to be extremely efficient in preventing COVID-19 infection in nonpregnant participants [6]. Because no clinical trials were conducted on pregnant or lactating women during the FDA approval process, whether the drugs are safe to use in these two populations has been questioned. Pfizer/BioNTech’s vaccination for pregnant women started its first phase of clinical trials on 20 February 2021, with the study’s findings yet to be disclosed [7]. Pregnant and breastfeeding women were excluded from the first round of anti-SARS-CoV-2 vaccine clinical trials due to fetal safety concerns, raising the question of when these patients will benefit from immunization.

After two doses of COVID-19 vaccination were administered to the general population, which consisted primarily of participants who had not previously been infected with SARS-CoV-2, there was a 90% efficacy in preventing severe symptoms for the COVID-19 variants that were dominant in 2020 and the first half of 2021. Following these encouraging findings, the American College of Obstetricians and Gynecologists, the Society for Maternal-Fetal Medicine, and the World Health Organization all endorsed the availability of COVID-19 vaccines for pregnant women beginning in 2021 through a shared decision process between the three organizations [8]. At the beginning of September 2021 in Romania, a decision was reached after discussions with medical societies and agencies about the risks and benefits of vaccination and the lack of available safety data. Preferential administration of the vaccine is recommended for pregnant women who are at high risk of severe infection [9]. By mid-December 2021, Romania registered 7.7 million people vaccinated with a complete scheme, BNT162b2 Pfizer leading with over 5 million individuals, while mRNA-1273 Moderna was used in just 400,000 people [10]. The Ad26.COV2.S vaccine was the second most used in the Romanian population, with over 1.9 million complete vaccinations. Similarly, pregnant women in Romania chose mainly BNT162b2, and secondly, the Ad26.COV2.S with only one dose.

SARS-CoV-2 spike protein, which is recognized by the immune system, is synthesized by the BNT162b2 vaccine, which is based on a unique technology that uses mRNA to generate the spike protein [11]. Pregnancy is contraindicated for several vaccines, including those that use live attenuated viruses. However, because this particular vaccine was not previously used in pregnant women before the COVID-19 pandemic, there is limited information on its efficacy, effectiveness, and safety in pregnancy. Although new studies debating this topic are emerging [12,13,14], the level of evidence remains low in researching whether this vaccine will induce immunity in pregnant women and whether it will impact the outcome of the pregnancy. The main reason for recommending the SARS-CoV-2 vaccine in pregnant women was the multitude of studies reporting more severe manifestations of COVID-19 during pregnancy and higher rates of preterm deliveries, thus justifying the use of mRNA vaccine in pregnancy [15].

Considering a significantly increased willingness among our pregnant patients in getting a COVID-19 vaccine after the official recommendations coming from ACOG, the International Federation of Gynecology and Obstetrics (FIGO), and the Romanian Health Ministry, we decided to develop a prospective study aiming to determine the immune response and adverse maternal and neonatal outcomes.

## 2. Materials and Methods

The study was designed as a prospective cohort to observe adult pregnant women who decided to vaccinate with the BNT162b2 and Ad26.COV2.S regardless of their SARS-CoV-2 infection status before or during pregnancy. All patients had a history of admission or investigations performed at the Obstetrics and Gynecology Clinic of the Timisoara Municipal Emergency Hospital. The research had a duration of 7 months, between 1 May 2021 and 1 December 2021, with an initial phase of 3 months allocated for patient recruitment, while the remaining 4 months represented the follow-up and data collection period. The research protocol was approved by the Ethics Committee of the “Victor Babes” University of Medicine and Pharmacy from Timisoara, Romania, and by the Ethics Committee of the Timisoara Municipal Hospital.

Being the largest Obstetrics and Gynecology hospital in Western Romania, our Obstetrics department averages approximately 300 live births per month. We identified a total of 906 pregnant women in their third trimester. Around 30% of them requested to be vaccinated for COVID-19 from 1 May 2021 until 1 August 2021, totaling 285 instances of full vaccination with BNT162b2 and Ad26.COV2.S. Considering our hospital serves a region with more than 400,000 women of reproductive age and a proportion of 0.9% live births [16] at a 30% vaccination rate in this population, we calculated an appropriate sample size of 297 vaccinated pregnant women. From the 285 vaccinated patients identified during the study period, 227 accepted to enroll in the study, out of which 53 had COVID-19 before or after enrollment. From the remaining 621 patients identified during the study period, 608 pregnant women were eligible to be studied, while 92 of them got a history of SARS-CoV-2 infection before or during pregnancy. To assess the vaccine’s effect on pregnancy outcomes in unvaccinated women, both trial groups (vaccinated and unvaccinated) did not analyze the pregnant women with present or prior SARS-CoV-2 infection. A schematic of the study development is described in Figure 1.

Our laboratory was equipped with the Elecsys Anti-SARS-CoV-2 assay that was used to detect antibodies against a recombinant protein representing the nucleocapsid antigen, whereas the Elecsys Anti-SARS-CoV-2-S-RBD assay was used to determine the concentration of antibodies against the SARS-CoV-2 spike protein receptor-binding domain (RBD). Both tests were conducted as directed by the manufacturer’s instructions, and the procedures utilized were based on a sandwich reaction involving two antigens. The tested serum was treated with biotinylated recombinant antigen-specific for SARS-CoV-2 or SARS-CoV-2-S-RBD and with ruthenium complex-labeled recombinant antigen-specific for SARS-CoV-2 or SARS-CoV-2-S-RBD. The photomultiplier Cobas e immunoassay analyzer was used to determine the chemiluminescent emission. Quantitative findings for the Elecsys Anti-SARS-CoV-2-S-RBD test for B.1.1.7 strain were obtained using the calibration curve created for the analyzer using two-point calibration and the calibration curve. A value of 0.80 U/mL was regarded as negative, whereas a concentration equal to or higher than 0.80 U/mL was considered positive. The approaches have a specificity and sensitivity of roughly 99% [17].

The study’s exposure criterion was the administration of the COVID-19 vaccination during the third pregnancy trimester with two separated doses by 21 days. Vaccination during the first trimester was discouraged throughout the research period, and those vaccinated during the second trimester did not have time to reach the delivery date. The main groups of variables and parameters assessed in our research comprised the background of pregnant women, pregnancy characteristics and complications, fetal characteristics and newborn complications, and spike antibody measurement.

The statistical analysis was carried out with IBM SPSS v.26 statistical programs. We computed the absolute and relative frequencies, the mean and standard deviation (SD), or median values and interquartile range (IQR) that were associated with each of the studied variables. For comparison of proportions, the chi-square and Fisher’s tests were employed, while for comparison of group differences in parametric and nonparametric data, the *t*-test and the Mann–Whitney *U*-test were used. The evolution of SARS-CoV-2 spike antibodies was quantified in seronegative and seropositive individuals using units per milliliter (U/mL). The data were presented as mean values and 95% confidence intervals. The difference in antibody titers across groups was determined using the Tukey multiple comparison test with testing correction. Overall, statistical significance was considered at α = 0.05.

## 3. Results

A total of 173 seronegative pregnant patients were compared with a group of 529 unvaccinated seronegative pregnant patients after receiving a second dose of BNT162b2 or one dose of Ad26.COV2.S after 24 weeks of pregnancy. Significant differences in background data between the two groups were registered in the age of patients and their place of origin. The vaccinated pregnant women had an average of 29.8 years old, compared with 31.2 years old in the unvaccinated group (*p*-value = 0.013), while the latter were significantly more prevalent living in rural areas from Romania (37.9% vs. 28.9%, *p*-value = 0.030). Patients’ pregnancy characteristics, the newborn characteristics, and their complications did not show any particular differences or abnormal findings (Table 1).

The 173 seronegative pregnant patients who received a COVID-19 vaccine in their third pregnancy trimester were evaluated for adverse events (Table 2), and their spike antibodies were assessed before and after vaccination for a 4-month duration to compare the profile of this group with 54 SARS-CoV-2 seropositive pregnant patients with the same characteristics. The observed findings regarding spike antibody profile and their graphical presentation are presented in Table 3 and Figure 2, respectively.

Seropositive pregnant women seemed significantly more affected by signs and symptoms, including fever (25.9% vs. 10.9%, *p*-value = 0.006) and lymphadenopathy (14.8% vs. 4.6%, *p*-value = 0.010) after completing the full vaccination scheme recommended at the time of the study. There were no severe adverse events in the study groups after receiving the COVID-19 vaccine. However, the spike antibody titers were statistically significantly higher in the seropositive group during the 4 months of sampling this laboratory analysis.

The 173 seronegative pregnant patients who received a COVID-19 vaccine in their third pregnancy trimester were case-matched by age and background on a 1 to 1 ratio with non-pregnant women. Table 4 and Figure 3 describe these findings. Compared with the previous analysis, seronegative pregnant and non-pregnant women did not show significant differences in their antibody titers after receiving the full vaccination scheme. However, we observed important differences in adverse events between pregnant and non-pregnant patients (Table 5). Non-pregnant patients encountered higher rates of adverse symptoms after vaccination, including myalgia (12.7% vs. 6.6%, *p*-value = 0.026), fever (16.7% vs. 10.1%, *p*-value = 0.039), and lymphadenopathy (10.5% vs. 4.8%, *p*-value = 0.022); although pregnant patients experienced fatigue more often (82.8% vs. 67.8%, *p*-value < 0.001). There were no severe events in these two groups after vaccination.

## 4. Discussion

In our cohort, we documented the serum antibodies after vaccine administration in all women. In a non-infected population, vaccination with mRNA and vector virus vaccines elicited strong spike antigen-specific IgG titers, although the serum levels decreased significantly over a 4-month duration. We demonstrated that immunization had no extra harmful effects in pregnant women compared to non-pregnant women. Additionally, some undesirable vaccination side effects were more prevalent in the group of seropositive pregnant women, than was in the control group, but of no important medical significance.

Other studies reported fever to occur more often in pregnant patients after vaccination, specifically in 3.7% after the first dose and 15.8% after receiving the second dose in Pfizer’s clinical study of the BNT162b2 mRNA vaccine. Although the absolute risk is minimal, fever during the first trimester of pregnancy is sometimes associated with an increased incidence of birth defects [18]. Although immunization in our study was not planned in the first trimester, our statistics demonstrating few pregnant patients developing fever after immunization are comforting. Additional findings and similar data were recently reported by other studies investigating the topic of COVID-19 vaccination in pregnant women [15,19]. The safety profile of BNT162b2 was described as insignificantly different after pregnant women were found to have equal rates of rash, fever, and fatigability after immunization as non-pregnant women [20,21]. Myalgia, arthralgia, and headache were significantly less common in pregnant women following each dose. At the same time, it was reported that local pain or swelling and axillary lymphadenopathy were significantly less common following the first and second doses, respectively. Paresthesia was significantly more common in the pregnant population following the second dose. As we opted for a cautious approach by studying the vaccination in the third trimester of pregnancy, other research reported that there were no significant differences in the rates of side effects among pregnant women according to whether the vaccine was given during the first, second, or third trimester, except for local pain/swelling, which was significantly less common following the first dose when given during the third trimester, and uterine contractions, which were significantly more common following the second dose when given during the third trimester [21].

Our study demonstrated that the BNT162b2 and Ad26.COV2.S vaccination resulted in a strong humoral response in the examined pregnant women. While seronegative pregnant women had significantly lower SARS-CoV-2 spike antibody titers than seropositive pregnant women after vaccination, the clinical implications of this finding are unknown. In contrast, other studies found no difference in vaccine-induced antibody titers, and there were no variations in antibody titers between vaccination trimesters [22]. These results should be interpreted cautiously, given the researchers examined two distinct types of SARS-CoV-2 vaccines and comprised a relatively limited number of individuals. It seems improbable that the difference in antibody levels reported in this research between pregnant and non-pregnant women after immunization is due to the groups’ significant age difference of 16 months, as other studies suggest [23].

Increased antibody levels may provide higher protection against variations capable of partly evading immunization. Our results indicate that spike antibody titers decline significantly in both pregnant and non-pregnant patients after receiving a full vaccination scheme of either BNT162b2 or Ad26.COV2.S. The same results occur in pregnant patients who were naturally immunized for SARS-CoV-2, although a significantly higher antibody level is produced when a COVID-19 vaccine is administered, thus offering longer protection against known variants. As a result, the clinical consequences of declining antibody levels after vaccination are unknown, and it is critical to identify S-antibody thresholds linked with protection against clinical outcomes. Thus, in light of recent recommendations in favor of booster vaccinations worldwide [24], and in light of the potentially rapid decline in spike antibody levels suggested by these data, heterologous regimens that elicit stronger antibody responses may provide more durable immunity and greater protection against emerging variants. Longer follow-up periods are required after administering booster doses in the pregnant population to determine the longevity of the immune response and the additional protection that might be offered for newer circulating SARS-CoV-2 variants.

The current study brings novelty to the topic of SARS-CoV-2 vaccination during pregnancy and describes important safety findings. Some of its strengths include the evaluation for adverse events, and monthly check of the S-antibody titers, including the assessment of previous SARS-CoV-2 infection before vaccination, that allowed us to rule them out from the main study group. However, the lack of funding did not allow for multiple antibody comparison, such as the IgA and IgM levels in seropositive patients, or generation of anti-S1/anti-N, IgG, IgA, IgM, and total Ig antibodies, as it would have been useful to indicate whole protection and neutralizing capacity. Our research is limited by the sample size that failed to meet the calculations for an optimal size based on the studied population. Moreover, we did not stratify the cohort by vaccine of choice (either BNT162b2 or Ad26.COV2.S) to assess whether the findings were mainly from BNT162b2 or Ad26.COV2.S, nor did we check cross-reactivity for other coronaviruses, except for the B.1.1.7 strain.

## 5. Conclusions

The adverse impact profile and short-term obstetric and neonatal outcomes of pregnant women who received the BNT162b2 and Ad26.COV2.S vaccinations at any stage of pregnancy are not concerning. Although the spike antibody titers constantly decrease and halve after just 4 months, and the IgG response is significantly higher in seropositive individuals, there is indubitable positive immunogenicity of the BNT162b2 and Ad26.COV2.S vaccine types against the most commonly circulating SARS-CoV-2 viral strains in the pregnant population.

## Figures and Tables

**Figure 1 viruses-14-00307-f001:**
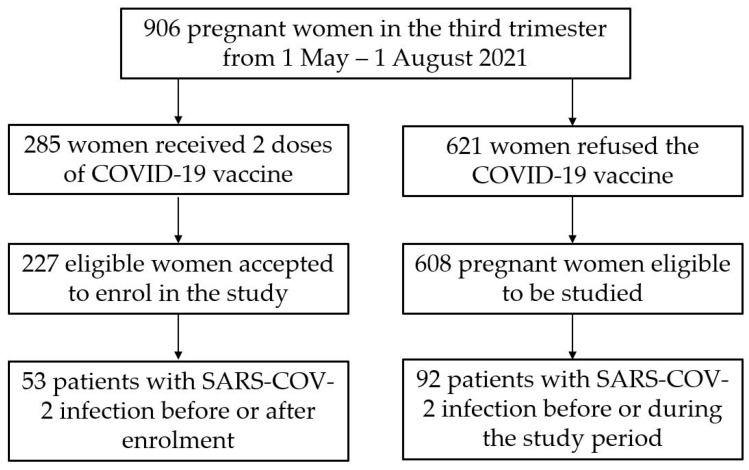
Flowchart of the study cohort. A total of 906 pregnant women were identified in their third trimester of pregnancy during the study period in the hospital database. Based on selection criteria, 285 women were eligible for inclusion in the vaccinated group after receiving two doses of COVID-19 vaccine, and the other 621 patients comprised the unvaccinated group. In total, 53 vaccinated pregnant women refused to consent for participation in the current study, leaving for a total of 227 eligible women in the vaccinated group. Moreover, 13 unvaccinated pregnant women refused to consent for participation, leaving for a total of 608 cases in the unvaccinated group. From the number of patients included in the study, those who suffered of SARS-CoV-2 infection (seropositive) before or after enrolment were separated from the main groups to stratify the data. A total of 53 vaccinated pregnant women and 92 unvaccinated pregnant women in their third trimester had a SARS-CoV-2 infection.

**Figure 2 viruses-14-00307-f002:**
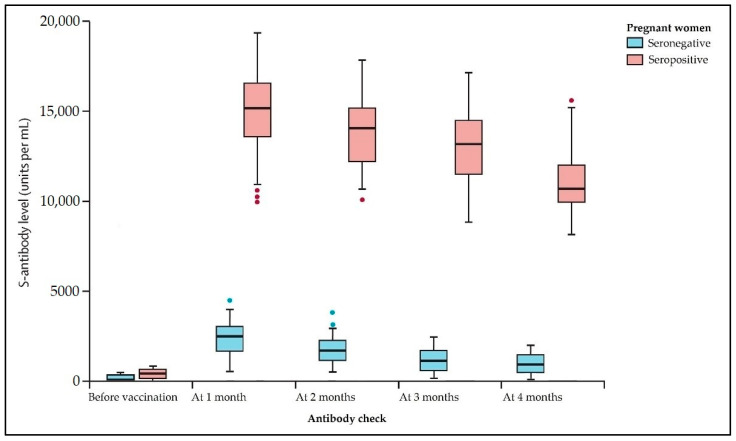
Boxplot comparison of spike antibodies in seronegative vs. seropositive pregnant patients. Data was evaluated in a seriated fashion, being followed before vaccination every month until 4 months. Median values and Interquartile Range are represented inside the box; minimum, maximum, and outliers are shown outside the box.

**Figure 3 viruses-14-00307-f003:**
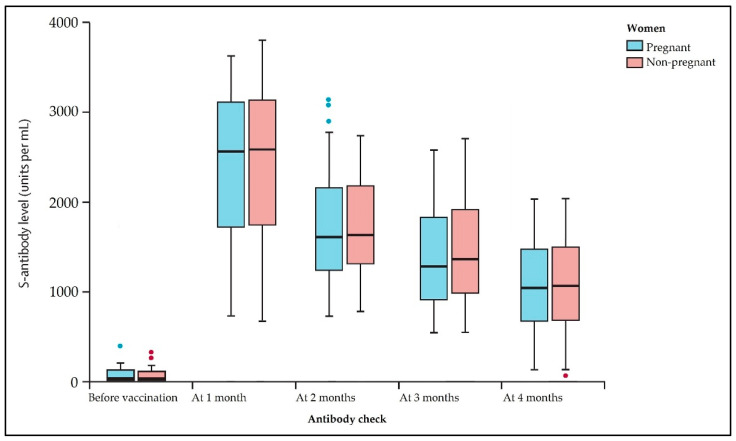
Boxplot comparison of pregnant vs. non-pregnant patients’ spike antibodies. Data was evaluated in a seriated fashion, being followed before vaccination every month until 4 months. Median values and interquartile range are represented inside the box; minimum, maximum, and outliers are shown outside the box.

**Table 1 viruses-14-00307-t001:** Comparison between vaccinated and unvaccinated pregnant women without COVID-19 history.

Variables *	Vaccinated (*n* = 173)	Unvaccinated (*n* = 529)	*p*-Value
**Background**			
Age, years (mean ± SD)	29.8 ± 6.1	31.2 ± 6.6	0.013
Obesity (BMI ≥ 30 kg/m^2^)	34 (19.6%)	119 (22.4%)	0.413
Infertility treatment	8 (4.6%)	21 (3.9%)	0.707
Rural place of origin	50 (28.9%)	201 (37.9%)	0.030
Unmarried	13 (7.5%)	58 (10.9%)	0.191
Unemployed	39 (22.5%)	148 (27.9%)	0.160
Multiparity	72 (41.6%)	215 (40.6%)	0.820
History of abortion	31 (17.9%)	86 (16.2%)	0.610
BNT162b2	115 (66.4%)	-	-
**Pregnancy characteristics and complications**			
No prenatal care	17 (9.8%)	74 (13.9%)	0.157
Gestational diabetes mellitus	12 (6.9%)	26 (4.9%)	0.307
Gestational hypertension	9 (5.2%)	15 (2.8%)	0.136
Oligohydramnios	6 (3.4%)	16 (3.0%)	0.771
Polyhydramnios	4 (2.3%)	15 (2.8%)	0.712
Abnormal presentation	7 (4.0%)	23 (4.3%)	0.864
Placental abruption	5 (2.9%)	18 (3.4%)	0.742
Assisted birth	7 (4.0%)	25 (4.7%)	0.709
Cesarean delivery	20 (11.5%)	69 (13.0%)	0.610
Preterm delivery	14 (8.1%)	37 (6.9%)	0.629
Postpartum hemorrhage	3 (1.7%)	7 (1.3%)	0.692
Endometritis	4 (2.3%)	14 (2.6%)	0.809
Hospital stay, days (median, IQR)	5 (1–10)	5 (1–14)	0.894
**Fetal characteristics and newborn complications**			
APGAR score <7 at 5 min	2 (1.1%)	5 (0.9%)	0.808
Abnormal fetal monitoring	11 (6.3%)	35 (6.6%)	0.905
Meconium aspiration	7 (4.0%)	31 (5.8%)	0.360
Small for gestational age	6 (3.4%)	26 (4.9%)	0.428
Weight, grams (mean ± SD)	3149 ± 380	3207 ± 362	0.071
Fever	2 (1.1%)	7 (1.3%)	0.865
ARDS	1 (0.6%)	5 (0.9%)	0.648
Hospital stay, days (median, IQR)	4 (1–14)	3 (1–19)	0.613

* Data reported as *n* (frequency) unless specified differently.

**Table 2 viruses-14-00307-t002:** Comparison of seronegative vs. seropositive pregnant patients’ adverse effects after receiving the COVID-19 vaccine.

Adverse Effects *	Seronegative (*n* = 173)	Seropositive (*n* = 54)	*p*-Value
Local pain	131 (75.7%)	35 (64.8%)	0.114
Arm numbness	13 (7.5%)	5 (9.2%)	0.678
Myalgia	10 (5.7%)	5 (9.2%)	0.368
Arthralgia	5 (2.8%)	2 (3.7%)	0.762
Fever	19 (10.9%)	14 (25.9%)	0.006
Fatigue	129 (74.5%)	37 (68.5%)	0.381
Lymphadenopathy	8 (4.6%)	8 (14.8%)	0.010
Rash	3 (1.7%)	2 (3.7%)	0.389
Headache	14 (8.1%)	5 (9.2%)	0.786
Severe events **	-	-	-

Seronegative = no history of SARS-CoV-2 infection; * dose independent; ** including anaphylaxis, thrombosis, Guillan–Barre syndrome, myocarditis, death.

**Table 3 viruses-14-00307-t003:** Comparison of spike antibodies in seronegative vs. seropositive pregnant patients.

Antibody Check *	Seronegative (*n* = 173)	Seropositive (*n* = 54)	*p*-Value
Before vaccination	0.41 (0.31–0.45)	145 (98.2–208.1)	<0.001
1 month	2433 (1752–3094)	15,360 (13,551–16,318)	<0.001
2 months	1697 (1393–2115)	14,571 (12,628–15,337)	<0.001
3 months	1314 (1095–1762)	12,870 (10,644–14,349)	<0.001
4 months	1083 (896–1468)	10,759 (9043–12,571)	<0.001

* Spike antibodies measured in U/mL; data presented as median (IQR).

**Table 4 viruses-14-00307-t004:** Case-matched comparison of pregnant vs. non-pregnant patients’ spike antibodies. Patients evaluated did not have a history of SARS-CoV-2 infection.

Antibody Check *	Pregnant (*n* = 173)	Non-Pregnant (*n* = 173)	*p*-Value
Before vaccination	0.41 (0.31–0.45)	0.40 (0.32–0.47)	0.457
1 month	2433 (1752–3094)	2461 (1719–3176)	0.627
2 months	1697 (1393–2115)	1705 (1452–2179)	0.823
3 months	1314 (1095–1762)	1377 (1147–1822)	0.059
4 months	1083 (896–1468)	1114 (909–1483)	0.352

* Spike antibodies measured in U/mL; Data presented as median (IQR).

**Table 5 viruses-14-00307-t005:** Case-matched comparison of pregnant vs. non-pregnant patients’ adverse effects after receiving the COVID-19 vaccine.

Adverse Effects *	Pregnant (*n* = 227)	Non-Pregnant (*n* = 227)	*p*-Value
Local pain	175 (77.1%)	162 (71.3%)	0.163
Arm numbness	18 (7.9%)	22 (9.6%)	0.507
Myalgia	15 (6.6%)	29 (12.7%)	0.026
Arthralgia	4 (1.7%)	7 (3.1%)	0.359
Fever	23 (10.1%)	38 (16.7%)	0.039
Fatigue	188 (82.8%)	154 (67.8%)	<0.001
Lymphadenopathy	11 (4.8%)	24 (10.5%)	0.022
Rash	4 (1.7%)	3 (1.3%)	0.703
Headache	14 (6.1%)	18 (7.9%)	0.463
Severe events **	-	-	-

* Dose independent; ** including anaphylaxis, thrombosis, Guillan–Barre syndrome, myocarditis, death.

## Data Availability

The data presented in this study are available on request from the corresponding author.

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
