# Peer review of "Immunogenicity Following Administration of BNT162b2 and Ad26.COV2.S COVID-19 Vaccines in the Pregnant Population during the Third Trimester"

_viruses, 2022, doi:10.3390/v14020307_

Round 1

Reviewer 1 Report

The manuscript entitled "Immunogenicity Following Administration of BNT162b2 and Ad26.COV2.SCOVID-19 Vaccines in the Pregnant Population During the Third Trimester" by Mihaela Citu et al. described the adverse events and the four-month immunogenicity results of pregnant women receiving the BNT162b2 and Ad26.COV2.S vaccinations, emphasizing the potent antigen-specific IgG response after immunizations, which is useful for promoting the vaccination campaign of pregnant women.

However, there are a few points the authors should consider in their modified manuscript:

  1. The authors include the participants who received BNT162b2 and Ad26.COV2.S vaccines, however, they do not really compared the specific adverse events and immune response (antibody concentrations) from each of them. This is also important for readers who would like to learn more details about the difference on vaccine efficacy of investigated two vaccines. For instance, the findings indicated the participants who developped lymphadenopathy, one would ask whether they were mainly from BNT162b2 or Ad26.COV2.S vaccinations?
  2. In this manuscript, only S-specific antibody concentrations were detected from the tested serum, those immunogenicity data are rather limited in term of evaluation of specific vaccines. They did not mention the SARS-CoV-2 strain information for Elecsys Anti-SARS-CoV-2 (-S-RBD) assays. Another important remark is the lack of neutralizing antibody test. As we kown the antibody binding titers do not indicate the neutralizing capacity of those antibody that confer more protective immunity to human.

Minor comments:

Line 139-140, it is unclear for the standard of negative/positive value.

“A value of 0.80 U/mL 139 was regarded as negative, whereas a concentration of 0.80 U/mL was seen as positive.”

Author Response

Dear reviewer,

We all appreciate your feedback and the time taken to evaluate our manuscript. In order to improve our paper, we made the following edits based on your advice, in addition to the other reviewers:

  1. We understand your position towards comparing the BNT162b2 and Ad26.COV2 vaccines by symptoms and immune response, and we also believe this would be an interesting research topic in pregnant women. However, our main focus was the reactions the COVID-19 vaccines are having for pregnant women, since the main regulatory bodies for this issue recommended vaccination in this population, although only preliminary data was available to them, not actual clinical trials. In fact, we also considered to include in our research the mRNA-1273 Moderna vaccine, but at the time of study we observed that an insignificant number of pregnant women chose that option, so, we were eventually left with the Pfizer and Jansen vaccines. In conclusion, we can consider performing another study that aims to study in particular these vaccines in the pregnant population, but if we were to add this additional data to the existent manuscript, it would reshape the whole study.

  1. Line 148: We specified the SARS-CoV-2 strain B.1.1.7 being analyzed for Elecsys Anti-SARS-CoV-2 test. It was the most prevalent at the time of study in Romania (summer of 2021). Unfortunately, besides analyzing the S-specific antibody concentrations, our study did not receive any funds to perform multiple tests. This was the most we were able to do with existent materials.

  1. We modified the figures’ legends (1, 2 and 3) to describe in detail the information they are meant to present.
  2. Lines 139-140 (Now line 150): equal or more than 0.80 U/mL was considered positive.
  3. Lines 229-230 (Now 246-247): we added the existent literature describing additional findings.
  4. References 19 and 20 were added.

Please let us know if you consider additional edits are mandatory.

Best regards,

The authors

Reviewer 2 Report

In the manuscript "Immunogenicity Following Administration of BNT162b2 and Ad26.COV2.S COVID-19 Vaccines in the Pregnant Population During the Third Trimester" the authors report a study performed on pregnant women and age-matched controls, who were vaccinated in the third trimester of pregnancy and analyzed antibody titers and adverse side effects for four months. They report that the vaccination was safe and induce comparable side effects in pregnant women as in the control group and that also the antibody titers were comparable. While the study is interesting, it is lacking novelty as already other studies about vaccination of pregnant women were published.

I have some minor revisions:

  • the figure legends are too short and missing important information about the presented data, performed statistics etc
  • Line 139-141: 0.8 U/ml is negative but also positive? There seems to be a mistake. The sentence should be checked
  • Line 228-229: Literature is missing for this sentence.

Author Response

Dear reviewer,

We all appreciate your feedback and the time taken to evaluate our manuscript. In order to improve our paper, we made the following edits based on your advice, in addition to the other reviewers:

  1. We understand your position regarding the novelty of the topic, although we still believe our study is relatively new if we consider the interval that data was analyzed from. Also, it is important for the population of pregnant women in Romania as proof of safety, and for being specific for the third trimester of pregnancy.

  1. We modified the figures’ legends (1, 2 and 3) to describe in detail the information they are meant to present.
  2. Lines 139-140 (Now line 150): equal or more than 0.80 U/mL was considered positive.
  3. Lines 229-230 (Now 246-247): we added the existent literature describing additional findings.
  4. References 19 and 20 were added.

Please let us know if you consider additional edits are mandatory.

Best regards,

The authors

Round 2

Reviewer 1 Report

I would suggest the authors combine their noted comments (point 1 and 2) to the limitations of their study (at the end of the "discussion" part of this manuscript).

Author Response

Dear reviewer,

Thank you again for your valuable feedback.

Based on your recommendation we decided to revise the last paragraph from Discussion, where we discussed the strengths and limitations, and why we did not stratify data by vaccine type, and the SARS-CoV-2 strain assessed.

Please see lines 288-300.

Best regards,

The authors